# Optical mapping of biological water in single live cells by stimulated Raman excited fluorescence microscopy

Lixue Shi [1], Fanghao Hu [1] & Wei Min [1]*

Water is arguably the most common and yet least understood material on Earth. Indeed, the biophysical behavior of water in crowded intracellular milieu is a long-debated issue. Understanding of the spatial and compositional heterogeneity of water inside cells remains elusive, largely due to a lack of proper water-sensing tools with high sensitivity and spatial resolution. Recently, stimulated Raman excited fluorescence (SREF) microscopy was reported as the most sensitive vibrational imaging in the optical far field. Herein we develop SREF into a water-sensing tool by coupling it with vibrational solvatochromism. This technique allows us to directly visualize spatially-resolved distribution of water states inside single mammalian cells. Qualitatively, our result supports the concept of biological water and reveals intracellular water heterogeneity between nucleus and cytoplasm. Quantitatively, we unveil a compositional map of the water pool inside living cells. Hence we hope SREF will be a promising tool to study intracellular water and its relationship with cellular activities.

[1] Department of Chemistry, Columbia University, New York, NY 10027, USA. *email: wm2256@columbia.edu

Water plays an imperative role in myriad cellular processes. How and to what extent intracellular water might behave differently from the bulk water is a long-debated subject with profound implications to biophysics and cell biology[1–6]. Increasingly more spectroscopic studies have revealed the existence of a certain portion of slow water in living cells[7–13] which might stem from noncovalent interactions with surrounding biomolecules. For example, perturbed slow water has been identified in monocellular organisms like *Escherichia coli*[7–10]. For mammalian cells, anomalous properties of cellular water compared to normal bulk have also been reported[11–13]. As such, a compelling concept of biological water has emerged, suggesting that intracellular water is different from bulk water at the molecular level. However, several fundamental questions about the heterogeneity of biological water are unanswered. First, the compositional nature of biological water is unclear. Second, it remains elusive as to how biological water is spatially distributed inside living cells. These questions are especially critical for mammalian cells in which cellular activities take place at specific subcellular compartments.

Powerful techniques are needed in order to address the spatial and compositional heterogeneity of biological water inside living cells. For instance, nonlinear IR spectroscopy has contributed to visualizing the nature of hydrogen bonding as well as liquid water dynamics[14,15]. Unfortunately, most established spectroscopy techniques are not sensitive enough to characterize water state with subcellular resolution. Among them, IR spectroscopy coupled with vibrational probes is a promising approach. Besides directly probing the OH-stretch vibration of water, solvent–solute interactions can be calibrated to correlate with the vibrational frequency of a particular chemical probe[16]. One notable example is the characterization of local electric field via the vibrational Stark effect (VSE)[17–19]. Among various vibrational probes, nitriles are found to be critically sensitive to the H-bonding environment[20,21], which make them excellent water reporters. However, the IR approach is impeded by low detectability, poor spatial resolution, and severe water attenuation. As a result, it is largely restricted on spectroscopic studies in concentrated solutions.

Herein we set out to develop a Raman-based water mapping technique for living cells. Raman scattering is a complementary way to IR absorption with high spatial resolution and is free from water background influence, rendering it a more suitable approach for cellular imaging[22]. Raman sensitivity has undergone rapid revolution in the past decades. With coherent Raman scattering techniques, including coherent anti-Stokes Raman scattering (CARS)[23] and stimulated Raman scattering (SRS)[24] microscopy, imaging speed and sensitivity are significantly improved compared to conventional Raman[25,26]. Moreover, electronic pre-resonance SRS further pushes the sensitivity to sub-μM[27–29]. Furthermore, the most recently reported stimulated Raman exited fluorescence (SREF) microscopy, which encodes the Raman feature into the excitation spectrum of fluorescence emission, has accomplished the long-thought-after goal of detecting single-molecule vibration without plasmonic enhancement[30,31]. By leveraging SREF's superb sensitivity and resolution, here we extend and develop SREF into a water-sensing tool, by further coupling with vibrational solvatochromism of environment-sensitive Raman mode. This technique allows us to directly visualize spatially resolved distribution of water states inside single mammalian cells.

## Results

**SREF spectral imaging.** Leveraging the vibrational specificity and superb sensitivity of SREF, we set out to employ SREF for mapping biological water. Briefly, in SREF microscopy, two spatially and temporally overlapped pulsed (picosecond pulse width is used here) laser beams (called pump and Stokes) firstly drive the population from ground state to vibrational excited state. For sufficient vibrational excitation efficiency, a delicate electronic pre-resonance scheme is implemented: the pump beam energy is close to but lower than the absorption energy of the SREF probe. A follow-up probe beam, which is also pulsed and synchronized with the pump/Stokes pulses, upconverts the vibrational population further to the electronic excited states from which the fluorescence is emitted (Fig. 1a). To simplify the laser configuration, we use the same pump beam as the probe beam. As shown in Fig. 1b, a tunable pump/probe beam and a fixed Stokes beam (1031.2 nm) are focused to the sample with a high-NA objective, and fluorescence emission is detected with a confocal setup, which blocks out-of-focus signal and confers optical sectioning.

With matched excitation of both vibrational and fluorescence states, a nitrile-bearing dye, Rhodamine 800 (Rh800), has been demonstrated as an excellent SREF probe with single-molecule sensitivity[30]. We will be using Rh800 in this study too. By tuning the pump beam wavelength across the nitrile Raman peak (while the Stokes beam is fixed), a narrow-band vibrational peak can be acquired in fluorescence excitation spectrum (Fig. 1c). The broad background is mainly due to anti-Stokes fluorescence which follows Boltzmann distribution[30]. Note that CARS background generated by pump and Stokes beams is blocked here by optical filters. To accurately determine the vibrational frequency, the SREF spectrum is fitted as the sum of a Voigt line (convolution of Gaussian and Lorentzian lineshape) and an exponential decay background (Fig. 1c, Supplementary Fig. 1). Background-subtracted and laser-deconvoluted Raman spectra are presented in Fig. 1d. Consequently, both the SREF peak frequency and the linewidth are recorded.

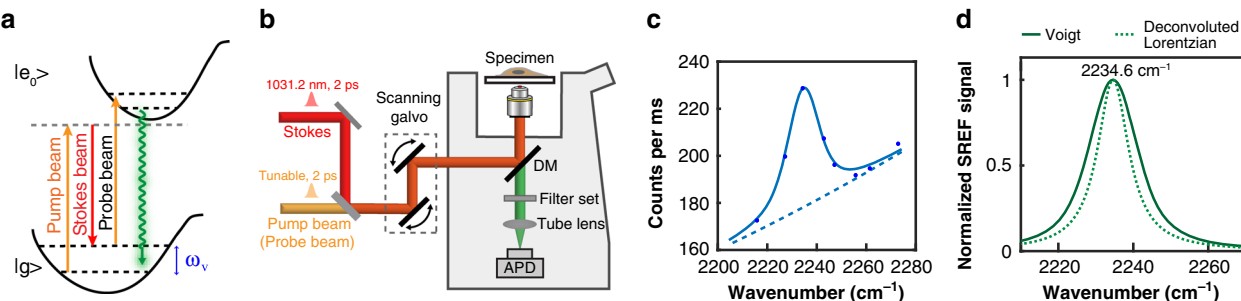

**Fig. 1** SREF microscopy. **a** Energy diagram for SREF process. **b** Microscope setup for SREF imaging. **c** SREF spectrum of 5-μM Rh800 in DMSO. SREF spectrum (solid line) is fitted with a sum of Voigt shape with an exponential decay background (dashed line). **d** Normalized SREF peak as Voigt shape (solid line) and Lorentzian shape after laser broadening deconvolution (dashed line)

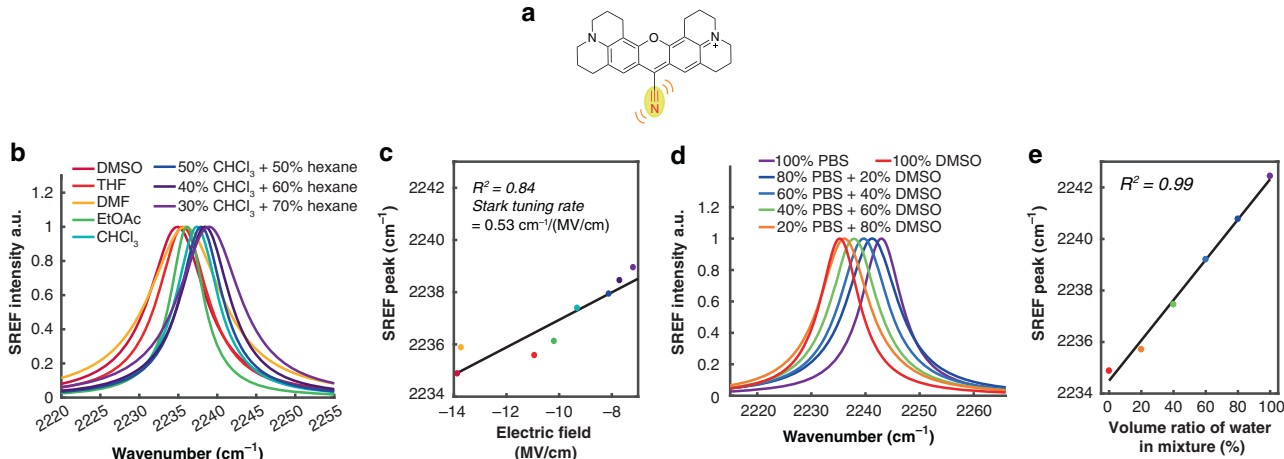

**Fig. 2** Spectroscopic characterization of vibrational sensing effect of SREF with Rh800. **a** SERF sensing with the nitrile mode of Rh800. **b** SREF spectra of C≡N band of Rh800 ($\bar{\nu}_{C\equiv N}$) dissolved in different aprotic solvents and mixtures. **c** $\bar{\nu}_{C\equiv N}$ follows a linear relationship with the electric field calculated from the Onsager reaction field theory. **d** SREF spectra of nitrile in different DMSO–water mixtures. **e** Plot of $\bar{\nu}_{C\equiv N}$ in DMSO–water binary mixture against water volume ratio

**Spectroscopic characterization on SREF vibrational sensing**. To establish and benchmark the sensing capability of SREF, we first studied vibrational solvatochromism of SREF under different solvents. The structure of our SREF probe Rh800 (Fig. 2a) contains a nitrile moiety, which is a well-studied vibrational probe in IR spectroscopy.

Linear Stark effect in which the frequency shift ($\Delta\bar{\nu}$) changes linearly with exerted electric field $\vec{F}$ as $hc\Delta\bar{\nu} = -\Delta\vec{\mu} \cdot \vec{F}$ (the coefficient value $|\Delta\vec{\mu}|$ is also known as the Stark tuning rate) dominates for vibrational transitions whose polarizability change is expected to be small[32]. In aprotic solvents without H-bonding, $\bar{\nu}_{C\equiv N}$ of Rh800 measured from SREF indeed shows redshift with the increase of solvent polarity (Fig. 2b, Supplementary Table 1), whereas no obvious trend could be concluded from the absorption spectra of Rh800 (Supplementary Fig. 2), indicating the exquisite sensitivity of vibrational mode to environment change. Moreover, $\bar{\nu}_{C\equiv N}$ follows a good linear relationship with the electric field calculated from the Onsager reaction field theory[33] (Fig. 2c, more details in Methods). The fitted $|\Delta\vec{\mu}|$, 0.53 ± 0.09 cm$^{-1}$/(MV/cm), is very close to the reported Stark tuning rate value of 0.61 cm$^{-1}$/(MV/cm) for aromatic nitriles in the IR spectroscopy studies[34]. Hence, we have established vibrational solvatochromism of the newly developed SREF spectroscopy of the nitrile-bearing probe.

Due to the strong H-bonding between nitrile and water, $\bar{\nu}_{C\equiv N}$ in aqueous environment would exhibit different behavior from that in aprotic non-H-bonding solvents (such as those in Fig. 2b), which complicates the analysis with electric field[16,35]. A reverse trend is usually observed with IR spectroscopy because of the opposite quadrupole interaction contribution besides the solvatochromic dipole interaction[16,36]. In accordance with the early IR spectroscopy work[20,34], we also observed a blueshift trend (opposite to the electrostatic effect, Supplementary Fig. 3) of the SREF peak for Rh800 in water (Fig. 2d, Supplementary Table 2). Quantitatively, our measured blue peak shift amount of 7.5 cm$^{-1}$ of Rh800 going from DMSO to water (Fig. 2d) matches well with the reported IR measurement of 7.8 cm$^{-1}$ on benzonitrile (an aromatic nitrile)[37], again supporting the agreement between Raman-based SREF spectroscopy and IR spectroscopy. Furthermore, we found that $\bar{\nu}_{C\equiv N}$ blueshifts with increasing water volume ratio in the water–DMSO binary mixture, which can be perfectly fitted with a linear relationship

(Fig. 2e). Together, these three experimental results support that $\bar{\nu}_{C\equiv N}$ measured in SREF of Rh800 could serve as a sensitive reporter of water (H-bonding) environment. It is worth noting that the H-bonding effect (Fig. 2e) induces a much larger frequency shift of Rh800 than the electrostatics effect (Fig. 2c).

**Water environment difference among single live HeLa cells**. With the water-sensing capability established, we then applied SREF spectral imaging in live HeLa cells (a human cancer-derived mammalian cell line), leveraging the high sensitivity and resolution of SREF. Without any cellular targets, Rh800 stains HeLa cells non-specifically. Higher concentration is used to obtain relatively homogeneous dye distribution inside cells. By sweeping the pump wavelength, a series of SREF images of Rh800-stained HeLa cells (Fig. 3a) are registered with no obvious photobleaching observed (Supplementary Fig. 4). After fitting peak frequency as in Fig. 1d, a $\bar{\nu}_{C\equiv N}$ distribution can then be generated (Fig. 3b). Note that this $\bar{\nu}_{C\equiv N}$ distribution is drastically different from the intensity-based SREF image (Fig. 3a), revealing new spectroscopy information beyond just the concentration distribution. Interestingly and importantly, the cellular distribution of $\bar{\nu}_{C\equiv N}$ reveals intriguing heterogeneity, with clearly higher $\bar{\nu}_{C\equiv N}$ inside nucleus than in cytoplasm (Fig. 3b, c). Moreover, this spatial heterogeneity is a robust feature that can be consistently observed in many cells (Fig. 3d, Supplementary Figs 5 and 6).

As a negative control to test the water effect, the $\bar{\nu}_{C\equiv N}$ distribution map in dried HeLa cells (whose water is removed by ethanol drying) reveals a relatively homogeneous pattern (Fig. 3e, f, Supplementary Fig. 6). Different from the bimodal distribution before (Fig. 3c, d, Supplementary Figs 5 and 6), only a single group is observed for the $\bar{\nu}_{C\equiv N}$ histogram of dried cells, which is consistent among multiple cells (Fig. 3g, h, Supplementary Fig. 7). The averaged $\bar{\nu}_{C\equiv N}$ in dried cells is 2238.1 cm$^{-1}$, which corresponds to the in vitro measurement in CHCl$_3$–hexane mixture. It is interesting to note that the dielectric constant of CHCl$_3$–hexane mixture is close to that reported for the folded protein[38], which is consistent with the expectation that the cellular interior after removing water is protein-dominated. Moreover, the variation of $\bar{\nu}_{C\equiv N}$ is less than 1 cm$^{-1}$ in dried cells, suggesting only mild electrostatics difference inside cellular environment (at the spatial resolution scale we are probing). Hence, this control experiment confirms that the heterogeneous

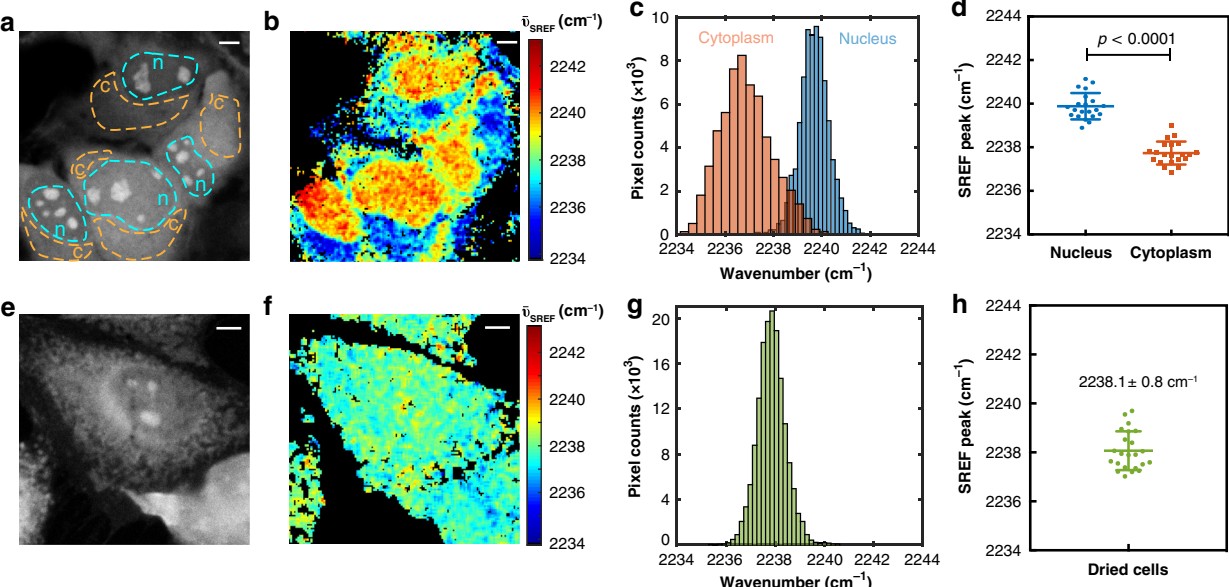

**Fig. 3** Mapping water heterogeneity in live cells. **a** A representative intensity-based SREF image of Rh800 stained living HeLa cells (pump = 836 nm). **b** Heterogeneous map of nitrile peak frequency. **c** Nitrile peak frequency histogram of two regions (n for nucleus and c for cytoplasm, marked in **c** with dash line) in live HeLa cells. **d** Nitriles peak frequency distribution (mean ± s.d.) of nucleus (2239.9 ± 0.6 cm$^{-1}$) and cytoplasm (2237.7 ± 0.5 cm$^{-1}$) among 21 HeLa cells. $p < 0.0001$ for two-sided paired $t$-test, $n = 21$. **e** A representative SREF image (pump = 836 nm) of ethanol-dried HeLa cells stained by Rh800. **f** Fitted frequency map shows the heterogeneities between nucleus and cytoplasm disappear for dried cells. **g** Nitrile peak frequency histogram of (**f**) indicating a single-group distribution. **h** Nitrile peak frequency distribution of 23 dried HeLa cells (mean ± s.d., 2238.1 ± 0.8 cm$^{-1}$). Scale bar: 5 μm. Error bars represent standard deviations

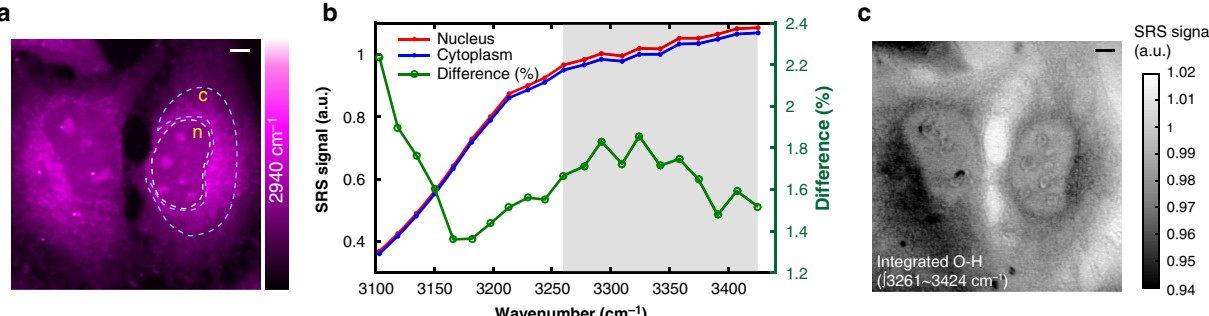

**Fig. 4** Water density inside live HeLa cells measured by SRS microscopy. **a** SRS image at frequency of 2940 cm$^{-1}$ (CH$_3$ stretching attributed mainly to protein) of live HeLa cells. **b** Spectral mapping of O–H stretching band at the nucleus and cytoplasm regions marked in (**a**) (n stands for nucleus and c stands for cytoplasm). The difference curve (shown on the right in percentage) is calculated as (nucleus−cytoplasm)/nucleus. **c** SRS image on integrated O–H stretching band over 3261–3424 cm$^{-1}$ (the gray area in (**b**), which is the spectral region that differs the most). Scale bar: 5 μm. The color scales display relative SRS signal level

distribution of $\bar{\nu}_{C\equiv N}$ we observed between nucleus and cytoplasm in vivo is from the water effect rather than the pure electrostatic effect.

**Water content is nearly constant inside live HeLa cells.** As suggested by Fig. 2e, the higher $\bar{\nu}_{C\equiv N}$ observed in nucleus than in cytoplasm indicates a more pronounced H-bonding environment. To explain such intracellular water heterogeneity, a straightforward interpretation would be a simple water content effect. In other words, one might speculate that nucleus may just have a substantially higher water concentration compared to the cytoplasm in HeLa cells. This hypothesis can be experimentally tested. Recently, by probing the O–H stretching mode with confocal Raman microscopy, the water density is reported to be only about 3% higher in nucleus than in the cytoplasm for live HeLa cells[39]. To revisit this issue, we then employed label-free SRS imaging of the O–H stretching mode, which has higher spatial resolution and

is free from sample auto-fluorescence compared to the previous confocal Raman approach (Fig. 4). A consistent result is obtained in our SRS imaging: nucleus exhibits about 2% higher water concentration than cytoplasm (Fig. 4b, c). Together, 2–3% water content difference between nucleus and cytoplasm will at most cause 0.2–0.3 cm$^{-1}$ frequency shift for $\bar{\nu}_{C\equiv N}$, given the full dynamic range of 7.5 cm$^{-1}$ in our calibration (Fig. 2e). Therefore, our observed large intracellular heterogeneity of $\bar{\nu}_{C\equiv N}$ cannot be explained by the simple water content variation.

Water environments include both the content and the state information. We noted that previous studies have suggested two types of water states, free water and biomolecule-bound water, existing in cells with distinct physical properties[2,40]. For biomolecule-bound water, the H-bonding capability to our Rh800 probe will be significantly weakened, which should lead to redshift of $\bar{\nu}_{C\equiv N}$. Inspired by this, we thus attributed higher $\bar{\nu}_{C\equiv N}$ in nucleus to a larger ratio of free water over bound water in

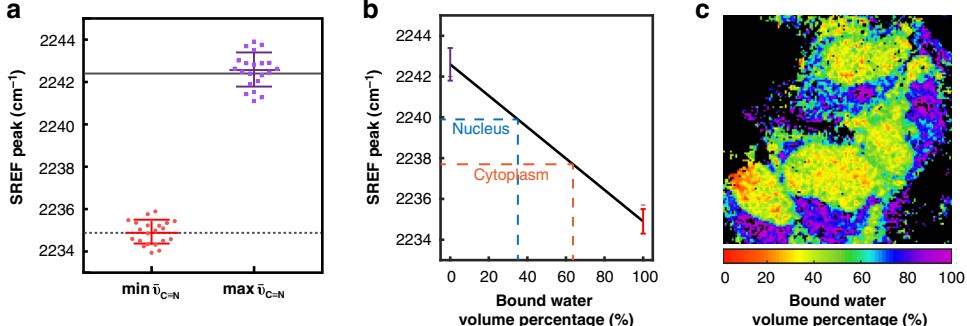

**Fig. 5** Compositional map of intracellular water pool. **a** Averaged minimum and maximum SREF peak frequencies of C≡N on Rh800 in cells. The minimum $\bar{\nu}_{C\equiv N}$ (mean ± s.d.) is 2234.9 ± 0.6 cm$^{-1}$ and maximum $\bar{\nu}_{C\equiv N}$ (mean ± s.d.) is 2242.6 ± 0.8 cm$^{-1}$ averaged over 21 HeLa cells. Solid line and dashed line show in vitro $\bar{\nu}_{C\equiv N}$ in pure water (2242.4 cm$^{-1}$) and DMSO (2234.9 cm$^{-1}$), respectively. **b** Linear calibration curve between bound water volume percentage and the SREF peak frequency. **c** Intracellular map for the bound-water state: 64% in cytoplasm and 35% in nucleus on average were estimated from **b**. Error bars represent standard deviations for 21 cells

nucleus than in cytoplasm. This qualitative trend indeed finds agreement with earlier spectroscopy results of faster solvation dynamics and more similarity to bulk-water dynamics inside nucleus than in cytoplasm[41–44]. Moreover, this attribution is also in the same trend with the concept of macromolecular crowding in which the cytoplasm is found to be somewhat more crowded than the nucleus[45,46].

**Compositional map of intracellular water pool**. To gain a more quantitative evaluation, we sought to convert the observed $\bar{\nu}_{C\equiv N}$ distribution to the spatially resolved compositional map of intracellular water pool. Specifically, inspired by the previous observation of the existence of free water and biomolecule-bound water, we aim to estimate their relative contribution to the water pool.

The statistics of $\bar{\nu}_{C\equiv N}$ collected from many cells reveal interesting insights. In particular, the maximum $\bar{\nu}_{C\equiv N}$ from many cells are rather similar to each other, and their average value (2242.6 ± 0.8 cm$^{-1}$) is very close to that in pure water (2242.4 cm$^{-1}$) in vitro (Fig. 5a), suggesting that it should represent the pure free water state. Similarly, the minimum $\bar{\nu}_{C\equiv N}$ of different cells also follows a narrow distribution around 2234.9 cm$^{-1}$. Interestingly, this number is very close to the frequency observed in very polar solvents such as DMSO. Note that the frequency shift can be approximated as the sum of the electrostatic effect (ranging from 2235 to 2239 cm$^{-1}$; Fig. 2c) plus the additional H-bonding effect (ranging 0–8 cm$^{-1}$; Fig. 2e). The only way the sum of these two effects being 2234.9 cm$^{-1}$ is that the H-bonding is vanishing and the local environment is as polar as DMSO (which is reasonable given the polar structure of water). This analysis thus prompts us to approximate the minimum $\bar{\nu}_{C\equiv N}$ as the bound-water state (Fig. 5a). Hence the blue and red boundaries of measured $\bar{\nu}_{C\equiv N}$ correspond to the two limits of free water and biomolecule-bound water, respectively.

After assuming a simple linear correspondence between the SREF peak frequency and the ratio between the free water and bound water (Fig. 5b), we can generate an intracellular map for the relative contribution from the free water state and the bound water state (Fig. 5c). From this map, we estimated the bound water composition to be 64% in cytoplasm and 35% in nucleus on average (Fig. 5b). When comparing our data to the literature, we note an interesting difference between the mammalian cell and simple organisms such as *E. coli* and yeast (in which the bound water composition was reported to be as low as ~20% estimated from time-resolved IR spectroscopy[9]): more bound water and less

free water in mammalian cells. Hence, our SREF-based water sensing provides a spatially resolved distribution of water states (bound vs free) inside single mammalian cells, offering the finest resolution to our best knowledge.

## Discussion

Our measured heterogeneity of SREF vibrational frequency (Fig. 3) can be potentially affected by three factors: the water content variation, the water states variation, and the electrostatics variation. In this regard, a model including all three terms shall be more general than our current model that only relates SREF frequency to the water states (Fig. 5b). Yet some limits can be estimated for each term to gain insights. Based on both the literature report and our own measurement, intracellular water density heterogeneity is a rather small effect (2–3%) in live HeLa cells (Fig. 4). Hence, the water content factor should not be the main contributor to our observation. Regarding the electrostatics effect, although we cannot exclude some special local cellular compartments with extreme electrostatics environments, this term is not likely to be dominating for nitrile bond which is known to be extremely sensitive to H-bonding[16,20,21,47]. After all, the frequency range (Fig. 2c) going from the most polar to the least polar aprotic solvents (that our probe can dissolve) is about half of the H-bonding effect (Fig. 2e)—the actual variation inside cells should be even smaller. Indeed, the dried cells exhibit relatively narrow distribution (about 0.8 cm$^{-1}$) of the SREF frequencies (Fig. 3h). Hence we believe this electrostatics variation is likely to be a minor effect when compared to H-bonding effect in live HeLa cells. Therefore, while we admit that the bound/free water state model might be oversimplified for more complex systems by omitting other factors (such as water content or electrostatics), its utility is justified by its tractability and quantitative outcome of spatially resolved distribution of water states (Fig. 5c).

Our result supported the concept of biological water, revealed intracellular water heterogeneity between nucleus and cytoplasm, and unveiled a compositional map of water pool (free versus bound water) inside living cells. We hypothesize that the water imbalance between cytoplasm and nucleus may attribute to their different functions. Cell cytoplasm contains numerous organelles, and a gel-like bath (i.e., more bound water) might facilitate close organelle interaction which is increasingly recognized to underlie many cellular functions[48]. In contrast, good fluidity (i.e., more free water) in nucleus matches with observed high mobility of nuclear proteins in literature[49] which might contribute to gene

expression activities. As a preliminary attempt of manipulating water pool, when we briefly block the water transfer with $HgCl_2$ (inhibitor for aquaporin), $\bar{\nu}_{C\equiv N}$ inside cytoplasm is found to redshift compared to the control (Supplementary Fig. 8), suggesting reduced free water. Meanwhile, we did not observe significant change of $\bar{\nu}_{C\equiv N}$ inside nucleus (Supplementary Fig. 8). This experiment shows that the nuclear envelope might play an active role in water transportation and could be consequential to maintaining water states inside nucleus under hazardous environments.

It is helpful to compare SREF with fluorescence and SRS, since SREF is an essentially a hybrid technique between these two. When comparing with SREF, either fluorescence or SRS alone is less ideal for this technical goal. Though fluorescence can achieve high sensitivity, the response is usually less informative due to the broad fluorescence spectrum and low sensitivity to the local H-bonding interaction (Supplementary Fig. 2). SRS needs to work on a higher probe concentration, which will introduce dye aggregation effect to complicate the interpretation. In fact, we found the aggregation effect will perturb SRS result but not SREF due to the extremely short excited state lifetime of the aggregates[50] (see Supplementary Fig. 9 and Supplementary Note 1 for details).

In summary, we have developed a promising imaging tool to spatially map biological water states in living cells, by coupling the local hydrogen-bond sensing ability of nitrile vibrational probe with the high sensitivity and resolution of the emerging SREF microscopy. This task will be very challenging for other techniques including fluorescence microscopy, infrared spectroscopy, or stimulated Raman scattering microscopy. Qualitatively we identified subcellular heterogeneity in live cells as observation of two distinct groups of biological water states between cytoplasm and nucleus. Quantitatively, we generated intracellular map of the water pool compositions regarding free water vs. bound water. Much of our results are direct visualization on single living cells. For future investigation, driving cells through key biological processes (such as cell division and apoptosis) and studying the corresponding changes of patterns in intracellular water pool might reveal interesting insight on intracellular hydration dynamics and cellular activity regulation.

## Methods

**SREF imaging system.** Synchronized pump and Stokes beams are provided by a picoEmerald S system from APE (Applied Physics & Electronic, Inc.) The fundamental IR fiber laser at 1031.2 nm with 2-ps pulse width and 80-MHz repetition rate is served as the Stokes beam. The pump beam is provided by incorporation part of fundamental laser to OPO which is tunable from 700 to 990 nm. Spatially overlapped pump and Stokes beams were expanded and coupled into an Olympus IX71 microscope to overfill the back-aperture of the objective by passing through a dichroic mirror (FF825-SDi01, Semrock). A 60× water immersion objective (1.2 N. A., Olympus UPLSAPO) was used for all imaging experiments. All laser power mentioned were measured after objective. Laser scanning was accomplished by a 2D galvo scanning system (GVSM002; Thorlabs). For SREF detection, fluorescence signal was detected on the backward with a high-efficient single photon counting module (SPCM) (SPCM-NIR-14-FC; Excelitas). The 100-μm active area diameter of avalanche photodetectors (APD) forms a loose confocal configuration. Two high-OD bandpass filters (FF01-729/167-25; Semrock) were used to block the reflected excitation laser beams and another two high-OD bandpass filter (FF01-735/28-15; Semrock) were utilized to block the CARS background. A home-written LabVIEW program was used to control the galvo scanning and data acquisition.

**SREF spectrum fitting and deconvolution of laser broadening.** The raw SREF spectra measured in this research can be interpreted as the sum of a Voigt lineshape (with fixed Gaussian pulse width of $9 \text{ cm}^{-1}$ laser broadening) and an exponential decay background (with both Boltzmann distribution of the fluorescence background and potential time accumulation of photobleaching effect considered). The Voigt profile is given as

$$V = \int_{-\infty}^{\infty} G(\bar{\nu}; \sigma) L(x - \bar{\nu}; \gamma) \mathrm{d}\bar{\nu}, \qquad (1)$$

where $x$ is the shift from the peak position; $G(\bar{\nu}; \sigma)$ is the Gaussian profile

describing laser broadening:

$$G(\bar{\nu}; \sigma) \equiv \exp\left(-\left(\frac{\bar{\nu} - \bar{\nu}_0}{\sigma/2\sqrt{\ln 2}}\right)^2\right). \qquad (2)$$

Here, $\bar{\nu}_0$ is the peak position, $\sigma$ is the laser pulse width (which is $9 \text{ cm}^{-1}$ in our case); and $L(\bar{\nu}; \gamma)$ is the Lorentzian profile (the deconvoluted SREF lineshape),

$$L(\bar{\nu}; \gamma) \equiv \frac{1}{1 + \left(\frac{\bar{\nu} - \bar{\nu}_0}{\gamma/2}\right)^2} \qquad (3)$$

$\gamma$ is the FWHM of the SREF peak. The integral can be related to Faddeeva function $\omega$ (also known as error function) as

$$V = \text{const.} \cdot \text{Re}[\omega(z)], \qquad (4)$$

where $\text{Re}[\omega(z)]$ is the real part of the Faddeeva function with

$$z = \frac{(\bar{\nu} - \bar{\nu}_0) + \mathrm{i}\frac{\gamma}{2}}{9 \text{ cm}^{-1}/2\sqrt{\ln 2}}. \qquad (5)$$

Therefore, the final fitting function will be

$$\text{raw SREF spectra} = x_1 \cdot \mathrm{e}^{x_2 \cdot (\bar{\nu} - 2264.3 \text{ cm}^{-1})} + x_3 \cdot \text{Re}[\omega(z)]. \qquad (6)$$

The five parameters $x_1, x_2, x_3, \bar{\nu}_0, \gamma$ contain all the information of the SREF lineshape, and will be numerically solved by fitting Eq. (6) to the raw 9-point SREF spectra measurements.

**Electric field estimated by Onsager reaction field theory.** In two different electrostatic environments with a difference in the electric field $\Delta\vec{F}$ (MV/cm), the shift in frequency for a vibrational probe is given by

$$hc\Delta\bar{\nu}_{\text{probe}} = \Delta\vec{\mu}_{\text{probe}} \cdot \Delta\vec{F}, \qquad (7)$$

where $h$ is Planck's constant; $c$ is the speed of light; $\Delta\vec{\mu}_{\text{probe}}$ is the dipole moment change between the ground and vibrational excited states; the value $|\Delta\vec{\mu}|$ is also known as the Stark tuning rate (in $\text{cm}^{-1}/(\text{MV/cm})$).

Onsager's reaction field theory considers the solute as a point dipole in a spherical cavity with continuous dielectric. The average solvent field is given by the Onsager reaction field[33]:

$$\left|\mathbf{F}_{\text{Onsager}}\right| = \frac{\mu_0}{a^3}\left[\frac{2(\varepsilon - 1)(n^2 + 2)}{3(2\varepsilon - n^2)}\right], \qquad (8)$$

where $n$ is the solute's refractive index and $a$ is the cavity radius; $\varepsilon$ is the dielectric constant of solvent; $\mu_0$ is the solute's gas-phase dipole moment in the ground state.

For Rh800, the refractive index could be estimated as $n = 1.5$ (similar to benzonitrile $n = 1.528$), since it should be highly insensitive to the electrostatic field. Due to the solubility issue for charged Rh800 molecule in nonpolar solvent, chloroform/hexane binary mixture were used as low polarity solvents. Dielectric constant for chloroform/hexane mixture is given by

$$\varepsilon_{\text{mixutre}} = \varepsilon_1 \cdot x_1 + \varepsilon_2 \cdot x_2, \qquad (9)$$

where $x$ is the molar fraction. Dipole moment was calculated by DFT calculations using B3LYP/6-31+G* basis set on Rh800. Gas phase dipole moment is calculated as 5.5978 Debye in the direction almost parallel to nitrile bond. Cavity radius $a$ was estimated from the molecule's density and formula weight. The density of Rh800 was estimated as $1.2 \text{ g/cm}^3$ (as for a similar dye Rhodamine 6G, $\rho = 1.26 \text{ g/cm}^3$).

**Sample preparation on Rh800-stained HeLa cells.** HeLa cells were first seeded on a glass-bottom dish with Dulbecco's modified Eagle's medium (DMEM) culture medium at 37 °C for 24 h. For live-cell experiments, cells were incubated with 2 μM Rh800 in DMEM medium for 30 min at 37 °C, then washed with phosphate-buffered saline (PBS) at RT before imaging. For dried-cell experiments, Cells were first incubated with 50 μM Rh800 in DMEM medium for 30 min at 37 °C. Then the cells were dried through ethanol drying by incubating with PBS, 10% EtOH in PBS, 20% EtOH in PBS, 50% EtOH in PBS, and 100% EtOH. The sample was finally dried in the vacuum desiccator before imaging.

**SREF spectra acquisition for solution and cell samples.** For the hyperspectral measurement on both solution and cell samples, the Stokes laser power was $P_{\text{Stokes}} = 10$ mW and the Pump laser power was $P_{\text{Pump}} = 12$ mW. Nine-point spectra were acquired by fixing the Stokes beam at 1031.2 nm and scanning the pump beam through 836–840 nm range with a 0.5-nm step. Solution samples were measured with 1-ms pixel dwell time. Cell images were acquired with 10-μs pixel dwell time and 100-nm pixel size. The acquisition time for a single image at a single frequency (500 × 500 pixels with 10-μs pixel dwell time) is about 4 s. Total spectral imaging for nine frequencies takes roughly 2 min.

**SREF peak mapping for cell images.** Equation (6) is used to numerically solve the SREF peak position with the 9-point hyperspectral image cube. The SREF peak of every 5 × 5-pixel region (pixel size = 100 nm) was generated by fitting of the sum of photons within the surrounding 11 × 11 pixel region. A mask was used for setting

all pixels with signal <8 photons to 0. To avoid possible local minimum in fitting, following criteria are applied for a good fit:

(1) FHWM ($\gamma$) is in the range of 10–30 cm$^{-1}$;
(2) interval variance of SREF peak with 90% confidence is smaller than 2.5 cm$^{-1}$;
(3) $x_3 > 0$ and $x_3 < 0.5 \cdot x_1$.

**Materials**. HeLa (ATCC CCL-2) cells were purchased from ATCC. Rhodamine 800 (Sigma 83701) and all solvents including hexanes, tetrahydrofuran (THF), dimethylformamide (DMF), chloroform, dichloromethane (DCM), acetone, and ethyl acetate were obtained from Sigma-Aldrich.

**Cell image statistics**. See Figs. 3d, 3h and 5a. Twenty-one live cells from 11 technical replicates and 23 dried cells from 8 technical replicates with each a 9-image set were used for statistical analysis. For data points in in Figs. 3d and 3h, pixel averaged SREF peak frequencies were calculated for single dried cells, or the cytoplasm and nucleus parts of single live cells. In Fig. 3d, the $p$ value is from a two-sided paired Student's $t$-test with $n = 21$.

**Reporting Summary**. Further information on research design is available in the Nature Research Reporting Summary linked to this article.

## Data availability
The data that support the findings of this study are available from the corresponding author upon reasonable request.

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

## Acknowledgements
We thank H. Xiong for his technical support on an SREF microscope. This work was supported by NIH R01 (GM128214), NIH R01 (GM132860), and the Camille and Henry Dreyfus Foundation.

## Author Contributions
L.S. and W.M. designed the project. L.S. performed the study, F.H. contributed to data interpretation, and all authors participated in paper writing.

## Competing Interests
The authors declare no competing interests.
