## [Peer Review File · Nature Communications]

Reviewers' comments:

Reviewer #1 (Remarks to the Author):

The manuscript NCOMMS-19-18336 reports a stimulated Raman excited fluorescence (SREF) microscopy study of HeLa cells. The authors use a nitrile based fluorescence dye in their study and map the resonance frequency of the CN stretching mode using SREF. Using different solvents and solvent mixtures, the authors demonstrate the sensitivity of the CN mode to the immediate environment of the fluorescent dye. Using this sensitivity, the authors map the CN mode in stained HeLa cells, revealing different resonance frequencies in the nucleus and the cytoplasm. Conversely, dried cells exhibit a more homogeneous distribution of CN resonance frequencies. The authors argue that the resonance frequency is a measure for free and bound water content in the cells and thus allows spatially resolving different water states in cells.

I find the approach and the experiments very innovative and the results are very interesting. The authors have studied and analysed their data very thoroughly and spatial mapping of the environment in living cells is certainly very timely. I however have some concerns about the interpretation of the data regarding the determination of free and bound water states. It appears to me that – rather than being sensitive to free and bound water – a more straightforward interpretation of the data would be that the CN frequency reports on the local water content in the cells. Thus, I can only recommend publication of the manuscript in Nature Communications if the authors provide further evidence for the sensitivity to bound water or if the authors revise their interpretation. Details follow below.

- My main concern is the experimental sensitivity to bound/free water: In Fig. 2b&c the authors show that the CN band is sensitive to the reaction field for non-hydrogen-bonding solvents with a full range of $\sim 5\text{cm}^{-1}$. For hydrogen-bonding solvents, the authors find the band shift to scale with water content in DMSO/water mixtures with a full range of $\sim 8\text{cm}^{-1}$. In cells, both effects are certainly at play and the hydrogen-bonding effect might prevail.

The scaling in Fig 2e however merely demonstrates that the band shift reports on the local water content as the center frequency exhibits a linear correlation with water volume fraction. In principle the findings of the authors would be consistent with a water content of 87% (2241cm^{-1} according to Fig 2e) in the nuclei and 44% (2237cm^{-1} according to Fig. 2e) in the cytoplasm as obtained from NMR experiments (Ref 38). While this sensitivity to water content can be supported by the data in Fig 2.

- The authors now use the CN frequency to obtain a free/bound water ratio. The authors arbitrarily define the lowest frequency, 2234.9 cm^{-1} , as the bound-water state and interpolate linearly starting from neat water. I do not find any compelling evidence for the experiments to be able to discriminate different water 'states', which is one of the major conclusions of this manuscript: (i) There is no obvious reason to assume that the lowest frequency observed corresponds to CN surrounded by bound water (it may well correspond to dyes embedded in membranes) and (ii) it leads to contradictory results (e.g. the dried cells, where I presume there is hardly any water left, would give $\sim 40\%$ of free water and in chloroform there would be also $\sim 40\%$ of free water according to this definition). Hence, resolving different water states, one of the major findings of this work, is somewhat questionable.

- Accordingly, it is also very dangerous to compare the thus obtained water fractions to NMR (ref 38) and infrared (ref 9) results as these experiments exclusively measure OH protons or OH stretching modes and use different correlation times to discriminate between bound and free water.

Altogether, I disagree with the notion that the experiments are sensitive to only water (and the subsequent decomposition into bound and free water), which is implied in the current version of the manuscript. The results however present a new and innovative means to look into the local environment in living cells. I believe the results are also without the focus on water sensing extremely exciting and could be published without such interpretation in Nat Commun.

Some more minor comments:

- It would be good to estimate the water content of the cells, in particular for the dried cells.
- The dye is ionic and the DMSO/water experiments are performed using a buffer solution. It would

be good if the authors could provide some information on the effect of the ionic strength on the CN frequency.

Reviewer #2 (Remarks to the Author):

This is an interesting manuscript in which the authors use the recently developed SREF technique for probing the intracellular environment. The nitrile moiety of Rh800 is used and a vibrational probe, which appears sensitive to the local environment. The sensitivity is attributed to the vibrational Stark effect and used here as a probe for the level of hydrogen bonding at different locations in the cell. From the Stark shift, the authors estimate a bond versus unbound water ratio, which they use to produce 'bound' water maps.

This work falls into the category of studies that use optical probes for sampling the local environment. For instance, different aspects of fluorescent probes have been used to monitor the intracellular environment, some of which are cited in the manuscript. The use of a vibrational probe is new, although one could argue that previous CARS and related work on intracellular water also uses a vibrational probe (namely the vibrational response of water itself). Nonetheless, the method outlined here is certainly new and the presentation of actual water maps is a step forward to what has been presented in the literature so far. Overall, the method is exciting and the results are interesting.

In order to improve their manuscript, I would like to point out the following:

- 1) In the introduction, the authors mention three questions about biological water. The first and third point appear very much the same, as they both refer to the spatial variation of biological water throughout the cell. The second point is also not clear. What do the authors mean by the composition of water? Are they referring to structural variation or chemical compositional variation? The chemical composition of water cannot vary, because water is just that: water!
- 2) There is a broad IR literature, mostly nonlinear optical measurements, that focuses on the nature of hydrogen bonding. The authors may want to make a reference or two to acknowledge this research field. Active groups include the ones led by Bakker and Woutersen.
- 3) Please add some experimental details: what is the estimated intracellular concentration of the probe? how fast is the spectral sweep performed? was there any evidence for photobleaching during the experiment?
- 4) The authors speculate that hydrogen bonding is responsible for the Stark shift. However, other local environment may cause a similar effect. For instance, the local ionic concentration can have a dramatic effect on the vibrational shift. Entropic effects may lead to congregation of specific ions near the dye in different parts of the cell, producing radically different shifts in different compartments. Also, how can the authors exclude the effects of acidity? The local proton concentration may affect the observations significantly. All in all, the hypothesis of bound versus unbound water is a nice guess, but I am not entirely convinced that other effects can be excluded.

Response to Reviewer #1

The manuscript NCOMMS-19-18336 reports a stimulated Raman excited fluorescence (SREF) microscopy study of HeLa cells. The authors use a nitrile based fluorescence dye in their study and map the resonance frequency of the CN stretching mode using SREF. Using different solvents and solvent mixtures, the authors demonstrate the sensitivity of the CN mode to the immediate environment of the fluorescent dye. Using this sensitivity, the authors map the CN mode in stained HeLa cells, revealing different resonance frequencies in the nucleus and the cytoplasm. Conversely, dried cells exhibit a more homogeneous distribution of CN resonance frequencies. The authors argue that the resonance frequency is a measure for free and bound water content in the cells and thus allows spatially resolving different water states in cells.

I find the approach and the experiments very innovative and the results are very interesting. The authors have studied and analysed their data very thoroughly and spatial mapping of the environment in living cells is certainly very timely. I however have some concerns about the interpretation of the data regarding the determination of free and bound water states. It appears to me that – rather than being sensitive to free and bound water – a more straightforward interpretation of the data would be that the CN frequency reports on the local water content in the cells.

Thus, I can only recommend publication of the manuscript in Nature Communications if the authors provide further evidence for the sensitivity to bound water or if the authors revise their interpretation. Details follow below.

First of all, we would like to thank the reviewer's deep appreciation of our work through his/her many positive comments such as "*I find the approach and the experiments very innovative and the results are very interesting. The authors have studied and analysed their data very thoroughly and spatial mapping of the environment in living cells is certainly very timely*".

We understand that the reviewer has concerns about the interpretation with bound/free water model. This was also pointed out by the reviewer as "*a more straightforward interpretation of the data would be that the CN frequency reports on the local water content in the cells*". We feel this is mainly due to the fact that we haven't thoroughly discussed this issue of water content. To elaborate on this issue, we have provided results from recent literature, conducted new experiments, and included a new Result section in the revised manuscript. As detailed below, we are confident that this concern has been thoroughly addressed in our revision. Meanwhile, more evidence for the sensitivity to bound water will also be provided below.

I however have some concerns about the interpretation of the data regarding the determination of free and bound water states. It appears to me that – rather than being sensitive to free and bound water – a more straightforward interpretation of the data would be that the CN frequency reports on the local water content in the cells.

We thank the reviewer for pointing out the issue of intracellular water content. Actually, water distribution inside living cells can be evaluated by Raman microscopy by probing the O-H stretching (**Fig. R1a**). A recent study (published in 2017) using confocal Raman imaging reported

that the water density is $\sim 3\%$ higher in nucleus than that in the cytoplasm for HeLa cells (**Fig. R1b-c**)¹. As shown in **Fig. R1c**, the intensity distribution of the nucleus is near unity (suggesting similar water density to cell medium) and the water density of cytoplasm is only $\sim 3\%$ smaller.

Fig. R1 Water density inside live HeLa cells measured by confocal Raman¹. (a) Raman spectrum of a HeLa cell. (b) Raman image of a single HeLa cell by plotting the O-H stretching band. (c) Relative intensity of the cytoplasm and nucleus of the integrated O-H stretching. Relative intensity is calculated by dividing the raw Raman intensity of nucleus or cytoplasm with the that of the cell medium. Figures are adapted with permission from *J. Phys. Chem. Lett.* 2017, 8, 21, 5241-5245. Copyright (2017) American Chemical Society.

To make sure this is indeed true, we decided to perform similar measurement by using hyperspectral Stimulated Raman Scattering (SRS) microscopy ourselves. Compared to confocal Raman, SRS imaging offers better spatial resolution and is free from autofluorescence background. **Fig. R2a** is SRS image at C-H stretching (2940 cm^{-1}) which visualizes the protein distribution of live HeLa cells, and cytoplasm and nucleus area can be identified from this image. By plotting the averaged SRS spectrum of pixels in nucleus and that in cytoplasm from a single live HeLa cell, we can see the relative signal difference is about 2% with nucleus containing slightly more water than cytoplasm (**Fig. R2b**). **Fig. R2c** shows water density map from the integrated Raman signal on part of O-H stretching peak ($3261\text{-}3424\text{ cm}^{-1}$, grey area in **Fig. R2b**). Therefore, our SRS microscopy measurement agrees well with the $2\sim 3\%$ intensity difference (**Fig. R1c**) reported in the literature on the same cell line by confocal Raman microscopy¹.

Fig. R2 Water density inside live HeLa cells measured by Stimulated Raman Scattering (SRS) in our lab. (a) SRS image at frequency of 2940 cm^{-1} (CH_2 stretching attributed mainly to protein) of live HeLa cells. (b) Spectral mapping on O-H stretching band at the nucleus and cytoplasm regions marked on Fig. R2a (n for nucleus and c for cytoplasm). The ‘difference’ curve is calculated as $(\text{nucleus}-\text{cytoplasm})/\text{nucleus}$. (c) SRS image on integrated O-H stretching.

Therefore, the important conclusion is that intracellular water density heterogeneity is a rather small effect based on both the literature report and our own measurement (Fig. R1-2). Quantitatively, 3% water density difference among nucleus and cytoplasm will at most cause 0.2-0.3 cm^{-1} frequency shift, considering a full dynamic range of about 8 cm^{-1} measured in our Fig. 2 E. This 0.2-0.3 cm^{-1} frequency shift is too small to explain our observed frequency difference in Fig. 3 C & D. In fact, this small shift is at the limit of our frequency measurement accuracy. This part of result and analysis has now been included in the revised manuscript as a new Fig 4.

- My main concern is the experimental sensitivity to bound/free water: In Fig. 2b&c the authors show that the CN band is sensitive to the reaction field for non-hydrogen-bonding solvents with a full range of $\sim 5 \text{ cm}^{-1}$. For hydrogen-bonding solvents, the authors find the band shift to scale with water content in DMSO/water mixtures with a full range of $\sim 8 \text{ cm}^{-1}$. In cells, both effects are certainly at play and the hydrogen-bonding effect might prevail. The scaling in Fig 2e however merely demonstrates that the band shift reports on the local water content as the center frequency exhibits a linear correlation with water volume fraction. In principle the findings of the authors would be consistent with a water content of 87% (2241 cm^{-1} according to Fig 2e) in the nuclei and 44% (2237 cm^{-1} according to Fig. 2e) in the cytoplasm as obtained from NMR experiments (Ref 38). While this sensitivity to water content can be supported by the data in Fig 2.

As the reviewer said, we agree hydrogen-bonding effect dominate our observation compared to pure electrostatic effect in cells, since CN bond is known to be extremely sensitive to hydrogen bonding. From the literature and our own SRS imaging result, the water content varies between nucleus and cytoplasm with only a 3% difference for HeLa cells. Therefore, we believe that water content distribution in HeLa cells is very different from that in X. oocytes (0.87 vs 0.44) obtained from NMR experiments in Ref 38.

As an experimental support for our method's sensitivity to water states, we also tested the CN band frequency dependence in aqueous solution with different concentration of Bovine Serum Albumin (BSA) protein. As shown in Fig. R3, the measurement indicates that ν_{CN} do red-shift with the increase of BSA concentration. As the solution volume doesn't change much after dissolving BSA powder, the water concentration is similar for different BSA solutions. On the other side, the bound water increase when BSA concentration increase as water forms H-bonds with BSA proteins. This in vitro test thus supports that CN band is sensitive to bound/free water.

Fig. R3 ν_{CN} of Rh800 red shift with increase of BSA concentration in DPBS buffer. Each point is calculated from three replicates of spectral sweeping.

We also note that the frequency shift is only 3 cm⁻¹ for BSA solution and tends to reach a plateau. This pure solution might be too simple to represent the true intracellular environments containing numerous types of proteins. Moreover, protein aggregation (e.g. dimerization²) could be substantial in high concentration of BSA (above 100 mg/mL). In contrast, the intracellular protein concentration (around 200 to 300 mg/mL³) is several times higher, likely due to the help of chaperons inside cells. Hence, one can expect more red shift inside *in vivo* cells which contain more concentration proteins as well as nucleic acids, lipids, sugars etc.

Finally, our attribution of higher $\bar{\nu}_{C\equiv N}$ in nucleus to a larger ratio of free water over bound water in nucleus than in cytoplasm is also in the same trend of various literature of different aspects. For example, earlier spectroscopy reported faster solvation dynamics and more similarity to bulk-water dynamics inside nucleus than in cytoplasm⁴⁻⁸. Moreover, this attribution is also in the same trend with the concept of macromolecular crowding in which the cytoplasm is found to be somewhat more crowded than the nucleus^{9,10}.

- The authors now use the CN frequency to obtain a free/bound water ratio. The authors arbitrarily define the lowest frequency, 2234.9 cm⁻¹, as the bound-water state and interpolate linearly starting from neat water. I do not find any compelling evidence for the experiments to be able to discriminate different water 'states', which is one of the major conclusions of this manuscript: (i) There is no obvious reason to assume that the lowest frequency observed corresponds to CN surrounded by bound water (it may well correspond to dyes embedded in membranes) and (ii) it leads to contradictory results (e.g. the dried cells, where I presume there is hardly any water left, would give ~40% of free water and in chloroform there would be also ~40% of free water according to this definition). Hence, resolving different water states, one of the major findings of this works, is somewhat questionable.

We thank the reviewer for this insightful question (especially question i; question ii is likely due to some misunderstanding). We now explain why using the lowest frequency of 2234.9 cm⁻¹ as the bound-water state is a reasonable assumption:

1) The lowest frequencies among different single cells collapse into a similar number, which implies that this number represents biological meaningful environment.

2) The measured frequency shift of the CN bond was determined by the electrostatic (i.e., non-H-bonding) effect and the additional H-bonding availability and as:

$$\nu_{CN} = \Delta\nu_{H-bonding} + \nu_{electrostatic}$$

From our *in vitro* calibration, we know $\nu_{electrostatic}$ lies in the range of 2235 to 2239 cm⁻¹ (Fig. 2 C) and $\Delta\nu_{H-bonding}$ is in the range of 0-8 cm⁻¹ (Fig. 2 E). The only way that the sum of these two terms to be around 2234.9 cm⁻¹ is that there is vanishing H-bonding contribution and the environment is as polar as DMSO. We thus hypothesize that this vanishing H-bonding condition should represent a bound-water state in which almost no H-bond can be formed with our CN probe. Meanwhile, the concurrent conclusion that bound-water is very polar (similar to DMSO) also makes chemical sense, considering the large difference of electronegativity of hydrogen and oxygen atom. In fact, the static dielectric constant of biological water was taken as 30 in previous work¹¹ which is not far from 47 of DMSO.

As to the second question, the Reviewer may have a misunderstanding on the outcome of H-bonding and pure electrostatic effect. We don't think the measurement result of dried cells and in chloroform are contradictory. Based on the literature of IR spectroscopic measurement on CN mode, a reverse trend is usually observed in H-bonded environment vs non-H-bonded environment due to the opposite quadrupole interaction contribution besides the solvatochromic dipole interaction^{12,13}. Because of this, the overall calibration curve between peak frequency and electric field will be V-shaped, and **Fig. R4a** depicts such a V-shaped curve adopted from the literature¹². Moreover, molecular dynamics simulated electric field and water percentage (v/v) in the DMSO-water mixture has a linear relationship for nitriles of both PhCN and MeSCN¹² (**Fig. R4b**). Incorporating these information, a similar V-shape calibration curve for SREF frequency can be drafted (**Fig. R4c**). For the dried cells, water has been completely removed by gradient ethanol exchange and drying (more details in **Fig.R5**). In such context, electrostatic effect rather than H-bonding effect determines the nitrile frequency, and one should refer to the right half of the curve derived from aprotic solvents. In other words, one shall not try to read out an effective water content by using the left half of the curve for dried cells or in chloroform.

Fig. R4 V-shaped SREF peak frequency-electric field correlation. (a) Field-frequency correlation curve for PhCN in H-bonding and non-H-bonding environments. Figure is adapted with permission from *J. Phys. Chem. B* 2016, 120, 17, 4034-4046 (<https://pubs.acs.org/doi/abs/10.1021/acs.jpcc.6b02732>). Further permissions related to the material excerpted should be directed to the ACS.

(b) Linear correlation for electric field and water percentage (v/v) in the DMSO-water mixture for nitriles of PhCN and MeSCN. MD estimated electric field data were from *J. Phys. Chem. B* 2016, 120, 4034-4046

(c) SREF peak frequency and field correlation as a reference curve for understanding peak differences among nucleus, cytoplasm of live cells and dried cells. The estimated electric field ($f \cdot F_{C\equiv N}$) for DMSO/water mixture was scaled from black curve (PhCN) in (b). Fit line for H-bonded condition is $\bar{\nu}_{C\equiv N} = -0.55 \cdot f \cdot F_{C\equiv N} + 2227$, ($R^2 = 0.99$). Fit line for Non-H-bonded condition is $\bar{\nu}_{C\equiv N} = 0.53 \cdot F_{C\equiv N, \text{Onsager}} + 2242.2$, ($R^2 = 0.84$).

- Accordingly, it is also very dangerous to compare the thus obtained water fractions to NMR (ref 38) and infrared (ref 9) results as these experiments exclusively measure OH protons or OH stretching modes and use different correlation times to discriminate between bound and free water.

We thank the reviewer for this point. We agree that the NMR or infrared results mainly rely on water dynamics to discriminate bound vs free water. We argue that one can certainly imagine that the H-bonding capability (which is what we are probing with SREF) is related to the apparent water dynamics, as slow water dynamics arises through forming H-bonding with surrounding biomolecules. Yet we agree that it is not clear how these two populations relate to each other quantitatively. An interesting future direction could be to correlate SREF with NMR/infrared measurements on the same sample. In the revision, we have pointed out this limitation explicitly.

Altogether, I disagree with the notion that the experiments are sensitive to only water (and the subsequent decomposition into bound and free water), which is implied in the current version of the manuscript. The results however present a new and innovative means to look into the local environment in living cells. I believe the results are also without the focus on water sensing extremely exciting and could be published without such interpretation in Nat Commun.

Again we sincerely appreciate Reviewer 1's strong support of our work. An important premise that water content variation is rather small in HeLa cell was unfortunately missing in our original manuscript. The inclusion of this new Result (Fig. 4) should help persuade the reviewer about the interpretation of our data.

Our measured heterogeneity of SREF vibrational frequency can be potentially affected by three factors: the water content variation, the water states variation and the electrostatics variation. As we have detailed above, intracellular water density heterogeneity is a rather small effect (2~3%) based on both the literature report and our own measurement (**Fig. R1-2**). Hence, this factor should not be the main contributor to our observation. Regarding the electrostatics effect, we argue that it is relatively small for CN bond which is known to be extremely sensitive to H-bonding. Although we cannot exclude some special local cellular compartments with extreme environments, this term is not likely to be dominating. As shown in **Fig. R4c**, the frequency range (4 cm^{-1}) of the right curve (even going from the most polar to the least polar aprotic solvents that our probe can dissolve) is about half of the left curve range (8 cm^{-1}) -- the actual variation inside cells should be smaller than 4 cm^{-1} . Indeed, the dried cells exhibit relatively narrow distribution (about 1 cm^{-1}) of the SREF frequencies (Fig. 3 H). Hence we believe this electrostatics variation is likely to be a minor effect when compared to H-bonding effect in live HeLa cells. However, for the purpose of keeping our interpretation more general, we have now added a new Discussion paragraph to explicit mention the limitation of our bound/free water model with a potential correction of the electrostatic term. Therefore, we hope the reviewer find our revision acceptable.

With all the new results, clarifications and revisions listed above, we hope the Reviewer 1 can now support the publication of our revised paper in Nature Comm. Overall, the many insightful comments from Reviewer 1 have helped a lot to improve the clarity of our revision!

Some more minor comments:

- It would be good to estimate the water content of the cells, in particular for dried cells.

For the dried cells, we dehydrated the cells with gradient ethanol solution. As shown in **Fig. R5**, O-H stretching band significantly decreased (more than 10 times) in ethanol-dried cells compared to Raman spectrum in **Fig. R1a**. A small bump around 3400 cm^{-1} could be some cross-talk from N-H stretching (with the peak at 3300 cm^{-1}) and O-H stretching from biomolecules. Therefore, we believe the water has been nearly completely removed for the dried cells.

Fig. R5 Raman spectrum of ethanol-dried HeLa cells.

- The dye is ionic and the DMSO/water experiments are performed using a buffer solution. It would be good if the authors could provide some information on the effect of the ionic strength on the CN frequency.

As shown in **Fig. R6** about ionic strength dependence, we found of ν_{CN} of our probe is insensitive to the ionic strength up to 350-mM NaCl. All frequency shifts are within 0.5 cm^{-1} .

Fig. R6 ν_{CN} of Rh800 is insensitive to ionic strength. Each point is calculated from three replicates of spectral sweeping in 0.01M buffer with different concentration of NaCl.

References

1. Takeuchi, M., Kajimoto, S. & Nakabayashi, T. Experimental Evaluation of the Density of Water in a Cell by Raman Microscopy. *J. Phys. Chem. Lett.* **8**, 5241-5245, (2017).
2. Levi, V. & González Flecha, F. L. Reversible fast-dimerization of bovine serum albumin detected by fluorescence resonance energy transfer. *Biochim. Biophys. Acta, Proteins Proteomics* **1599**, 141-148, (2002).
3. Ellis, R. J. Macromolecular crowding: an important but neglected aspect of the intracellular environment. *Curr. Opin. Struct. Biol.* **11**, 114-119, (2001).
4. Bowtell, R. W. *et al.* NMR microscopy of single neurons using spin echo and line narrowed 2DFT imaging. *Magn. Reson. Med.* **33**, 790-794, (1995).
5. Hsu, E. W., Aiken, N. R. & Blackband, S. J. Nuclear magnetic resonance microscopy of single neurons under hypotonic perturbation. *Am. J. Physiol., Cell Physiol.* **271**, C1895-C1900, (1996).
6. García-Martín, M. a. L., Ballesteros, P. & Cerdán, S. The metabolism of water in cells and tissues as detected by NMR methods. *Prog. Nucl. Magn. Reson. Spectrosc.* **39**, 41-77, (2001).
7. Sasmal, D. K., Ghosh, S., Das, A. K. & Bhattacharyya, K. Solvation dynamics of biological water in a single live cell under a confocal microscope. *Langmuir* **29**, 2289-2298, (2013).
8. Päufer, S., Zschunke, A., Khuen, A. & Keller, K. Estimation of water content and water mobility in the nucleus and cytoplasm of *Xenopus laevis* oocytes by NMR microscopy. *Magn. Reson. Imaging* **13**, 269-276, (1995).
9. Guigas, G., Kalla, C. & Weiss, M. The degree of macromolecular crowding in the cytoplasm and nucleoplasm of mammalian cells is conserved. *FEBS Lett.* **581**, 5094-5098, (2007).
10. Murade, C. U. & Shubeita, G. T. A Molecular Sensor Reveals Differences in Macromolecular Crowding between the Cytoplasm and Nucleoplasm. *ACS Sens.* **4**, 1835-1843, (2019).
11. Nandi, N. & Bagchi, B. Dielectric Relaxation of Biological Water. *J. Phys. Chem. B* **101**, 10954-10961, (1997).
12. Deb, P. *et al.* Correlating nitrile IR frequencies to local electrostatics quantifies noncovalent interactions of peptides and proteins. *J. Phys. Chem. B* **120**, 4034-4046, (2016).
13. Choi, J.-H. & Cho, M. Vibrational solvatochromism and electrochromism of infrared probe molecules containing C=O, C≡N, C=O, or C-F vibrational chromophore. *J. Chem. Phys.* **134**, 154513, (2011).

Response to Reviewer #2

This is an interesting manuscript in which the authors use the recently developed SREF technique for probing the intracellular environment. The nitrile moiety of Rh800 is used and a vibrational probe, which appears sensitive to the local environment. The sensitivity is attributed to the vibrational Stark effect and used here as a probe for the level of hydrogen bonding at different locations in the cell. From the Stark shift, the authors estimate a bound versus unbound water ratio, which they use to produce 'bound' water maps.

This work falls into the category of studies that use optical probes for sampling the local environment. For instance, different aspects of fluorescent probes have been used to monitor the intracellular environment, some of which are cited in the manuscript. The use of a vibrational probe is new, although one could argue that previous CARS and related work on intracellular water also uses a vibrational probe (namely the vibrational response of water itself). Nonetheless, the method outlined here is certainly new and the presentation of actual water maps is a step forward to what has been presented in the literature so far. Overall, the method is exciting and the results are interesting.

We are very encouraged by the Reviewer 2's commendations, and we appreciate the Reviewer's support. Below we will address reviewer's technical questions point-by-point.

In order to improve their manuscript, I would like to point out the following:

1) In the introduction, the authors mention three questions about biological water. The first and third point appear very much the same, as they both refer to the spatial variation of biological water throughout the cell. The second point is also not clear. What do the authors mean by the composition of water? Are they referring to structural variation or chemical compositional variation? The chemical composition of water cannot vary, because water is just that: water!

We thank the reviewer for raising this point. Yes, chemical composition of water is the same. Here we are referring to the structural variations like free water and bound water with different dynamics and H-bonding capability. Indeed, we put the spatial heterogeneity and structural heterogeneity as two dimensions which leads to three points we raised in the introduction part. That being said, we understand there might be some confusion here, and we have now modified the sentences to combine the first and third point as a general spatial variation of water property.

2) There is a broad IR literature, mostly nonlinear optical measurements, that focuses on the nature of hydrogen bonding. The authors may want to make a reference or two to acknowledge this research field. Active groups include the ones led by Bakker and Woutersen.

We thank reviewer for this constructive suggestion. We now add two spectroscopic studies on hydrogen bonding and water dynamic in our revised manuscript with an acknowledgement on nonlinear IR spectroscopy field. They are:

Woutersen, S., Emmerichs, U. & Bakker, H. J. Femtosecond Mid-IR Pump-Probe Spectroscopy of Liquid Water: Evidence for a Two-Component Structure. *Science* **278**, 658, (1997).

Woutersen, S. & Bakker, H. J. Hydrogen Bond in Liquid Water as a Brownian Oscillator. *Physical Review Letters* **83**, 2077-2080, (1999).

3) Please add some experimental details: what is the estimated intracellular concentration of the probe? how fast is the spectral sweep performed? was there any evidence for photobleaching during the experiment?

Estimated intracellular concentration:

We usually incubate the cells with 1-2 μM dye for 30 min. Based on the *in vitro* solution measurement in PBS buffer, we estimate the intracellular dye concentration will be around 10~100 μM on average.

How fast is the spectral sweep performed?

The acquisition time for a single image at single wavelength (e.g. 500x500 pixels with 10 μs pixel dwell time) will take about 4 seconds, and one round of spectral sweep (9 wavelength points) will need about 2 minutes.

Evidence for photobleaching:

The photobleaching effect is relatively negligible during SREF spectral sweeping, since rather low laser power and short pixel dwell time were used. For our experiments, we usually use 10-mW pump beam and 12-mW Stokes beam with 10 μs pixel dwell time. Moreover, both beams are not directly resonant with the dye absorption, as SREF employs a pre-resonance excitation condition. As illustrated in **Fig. R7**, two rounds spectral sweeping on same field of view of live HeLa cells gave reproducible readouts and the imaging pattern is also very consistent.

Fig. R7 Photobleaching characterization during SREF spectral sweeping. Two rounds of 9-point SREF spectral mapping were acquired with $P_{\text{pump}}=10$ mW and $P_{\text{Stokes}}=12$ mW with 10- μs pixel dwell time. Pump beam wavelength for each image is given at the lower left corner.

4) The authors speculate that hydrogen bonding is responsible for the Stark shift. However, other local environment may cause a similar effect. For instance, the local ionic

concentration can have a dramatic effect on the vibrational shift. Entropic effects may lead to congregation of specific ions near the dye in different parts of the cell, producing radically different shifts in different compartments. Also, how can the authors exclude the effects of acidity? The local proton concentration may affect the observations significantly. All in all, the hypothesis of bound versus unbound water is a nice guess, but I am not entirely convinced that other effects can be excluded.

Thanks for reviewer's question on these control experiments. **Fig. R8** is our experimental result. ν_{CN} of Rh800 is both insensitive to the ionic strength up to about 350 mM NaCl and the pH of broad range from 4-11. All frequency shifts are within 0.5 cm^{-1} .

Fig. R8 ν_{CN} of Rh800 is insensitive to both ionic strength and pH value. (a) Each point is calculated from three replicates of spectral sweeping in 0.01M NaPi buffer with different concentration of NaCl. (b) Each point is calculated from three replicates of spectral sweeping in DPBS buffer with different pH.

Our measured heterogeneity of SREF vibrational frequency can be potentially affected by three factors: the water content variation, the water states variation and the electrostatics variation. As we have detailed above, intracellular water density heterogeneity is a rather small effect (2~3%) based on both the literature report and our own measurement (**Fig. R1-2**). Hence, this factor should not be the main contributor to our observation. Regarding the electrostatics effect, we argue that it is relatively small for CN bond which is known to be extremely sensitive to H-bonding. Although we cannot exclude some special local cellular compartments with extreme environments, this term is not likely to be dominating. As shown in **Fig. R4c**, the frequency range (4 cm^{-1}) of the right curve (even going from the most polar to the least polar aprotic solvents that our probe can dissolve) is about half of the left curve range (8 cm^{-1}) -- the actual variation inside cells should be smaller than 4 cm^{-1} . Indeed, the dried cells exhibit relatively narrow distribution (about 1 cm^{-1}) of the SREF frequencies (Fig. 3 H). Hence we believe this electrostatics variation is likely to be a minor effect when compared to H-bonding effect in live HeLa cells. However, for the purpose of keeping our interpretation more general, we have now added a new Discussion paragraph to explicit mention the limitation of our bound/free water model with a potential correction of the electrostatic term. Therefore, we hope the reviewer find our revision acceptable.

Finally, our attribution to free vs. bound water is also in the same trend of various literature of different angles. For example, earlier spectroscopy reported faster solvation dynamics and more similarity to bulk-water dynamics inside nucleus than in cytoplasm¹⁻⁵. Moreover, this attribution is also in the same trend with the concept of macromolecular crowding in which the cytoplasm is found to be somewhat more crowded than the nucleus^{6,7}.

References

1. Bowtell, R. W. *et al.* NMR microscopy of single neurons using spin echo and line narrowed 2DFT imaging. *Magn. Reson. Med.* **33**, 790-794, (1995).
2. Hsu, E. W., Aiken, N. R. & Blackband, S. J. Nuclear magnetic resonance microscopy of single neurons under hypotonic perturbation. *Am. J. Physiol., Cell Physiol.* **271**, C1895-C1900, (1996).
3. García-Martín, M. a. L., Ballesteros, P. & Cerdán, S. The metabolism of water in cells and tissues as detected by NMR methods. *Prog. Nucl. Magn. Reson. Spectrosc.* **39**, 41-77, (2001).
4. Sasmal, D. K., Ghosh, S., Das, A. K. & Bhattacharyya, K. Solvation dynamics of biological water in a single live cell under a confocal microscope. *Langmuir* **29**, 2289-2298, (2013).
5. Päufer, S., Zschunke, A., Khuen, A. & Keller, K. Estimation of water content and water mobility in the nucleus and cytoplasm of *Xenopus laevis* oocytes by NMR microscopy. *Magn. Reson. Imaging* **13**, 269-276, (1995).
6. Guigas, G., Kalla, C. & Weiss, M. The degree of macromolecular crowding in the cytoplasm and nucleoplasm of mammalian cells is conserved. *FEBS Lett.* **581**, 5094-5098, (2007).
7. Murade, C. U. & Shubeita, G. T. A Molecular Sensor Reveals Differences in Macromolecular Crowding between the Cytoplasm and Nucleoplasm. *ACS Sens.* **4**, 1835-1843, (2019).

REVIEWERS' COMMENTS:

Reviewer #1 (Remarks to the Author):

The authors have answered most of my questions adequately. The authors have provided many arguments that support their interpretation. Despite I am not yet 100% convinced that the probe reports only on bound/free water, in the discussion now all the potential caveats are discussed. Thus, I recommend publication of the manuscript. I only have two minor comments/suggestions.

- I indeed overlooked the opposite sign of the slopes of the 'electrostatic' and the 'H-bonding' effects in Figure 2. To avoid any potential confusion, I recommend the authors to multiply the x axis of Figure 2c by -1. Alternatively, the authors could use a representation like in Figure R4 of their response to avoid potential confusion.

- I am still a bit puzzled about the comparison of the water content from this study (and ref 38) to the NMR results of ref 44. According to ref 44 the water content in the cytoplasm and the nucleus differs by nearly 100%, whereas Raman imaging finds only minor differences.

Given that the authors argue on the bottom of page 7 that the NMR determined bound water fractions are in line with their findings, I would find it fair to also mention in the section 'Water content is nearly constant inside single live HeLa cells' that here there is substantial disagreement with NMR. Possibly, conceivable origins of this discrepancy should be discussed (Could it be that Raman imaging is also sensitive to OH groups of biomolecules, while in NMR these peaks are very broad and only water gives rise to motionally narrowed signals?).

Reviewer #2 (Remarks to the Author):

The revised manuscript and the authors' response to my questions have addressed most of my previous concerns. I have no further comments.

Response to Reviewer #1

The authors have answered most of my questions adequately. The authors have provided many arguments that support their interpretation. Despite I am not yet 100% convinced that the probe reports only on bound/free water, in the discussion now all the potential caveats are discussed. Thus, I recommend publication of the manuscript. I only have two minor comments/suggestions.

Again, we want to thank Reviewer 1's support and all constructive suggestions which help clarify the paper more. To address the two remaining comments from Reviewer 1, we add a new supplementary Fig.3 (similar to Figure R4 in the last rebuttal) to emphasize the opposite trends of electrostatic and H-bonding effects. Moreover, we also delete the original ref44 and related discussion to avoid any unnecessary misleading.

- I indeed overlooked the opposite sign of the slopes of the 'electrostatic' and the 'H-bonding' effects in Figure 2. To avoid any potential confusion, I recommend the authors to multiply the x axis of Figure 2c by -1. Alternatively, the authors could use a representation like in Figure R4 of their response to avoid potential confusion.

As suggested by the reviewer, we now add V-shape representation like in Figure R4 as a new Supplementary Fig.3. Moreover, on page 4 in the main text, we emphasize the blue-shift trend in water is opposite to the solvatochromism trend when citing this new Supplementary Fig.3.

- I am still a bit puzzled about the comparison of the water content from this study (and ref 38) to the NMR results of ref 44. According to ref 44 the water content in the cytoplasm and the nucleus differs by nearly 100%, whereas Raman imaging finds only minor differences. Given that the authors argue on the bottom of page 7 that the NMR determined bound water fractions are in line with their findings, I would find it fair to also mention in the section 'Water content is nearly constant inside single live HeLa cells' that here there is substantial disagreement with NMR. Possibly, conceivable origins of this discrepancy should be discussed (Could it be that Raman imaging is also sensitive to OH groups of biomolecules, while in NMR these peaks are very broad and only water gives rise to motionally narrowed signals?).

We believe HeLa cells and X. oocyte are different biological systems. It is plausible that their intracellular water content distribution is not the same. In fact, the point we are making when citing ref44 on page 7 is rather minor. Therefore, in order to avoid any unnecessary misleading, we now decide to remove ref44 and its citation on page 7 all together.